# Retrospective Analysis Revealed an April Occurrence of Monkeypox in the Czech Republic

**DOI:** 10.3390/v14081773

**Published:** 2022-08-15

**Authors:** Martin Chmel, Oldřich Bartoš, Hana Kabíčková, Petr Pajer, Pavla Kubíčková, Iva Novotná, Zofia Bartovská, Milan Zlámal, Anna Burantová, Michal Holub, Helena Jiřincová, Alexander Nagy, Lenka Černíková, Hana Zákoucká, Jiří Dresler

**Affiliations:** 1Military Health Institute, Military Medical Agency, 16200 Prague, Czech Republic; 2Department of Infectious Diseases, First Faculty of Medicine, Charles University and Military University Hospital Prague, 12108 Prague, Czech Republic; 3National Reference Laboratory for Influenza and Respiratory Viruses, National Institute for Public Health, 10042 Prague, Czech Republic; 4State Veterinary Institute, 16503 Prague, Czech Republic; 5Department of Sexually Transmitted Infections, National Institute for Public Health, 10042 Prague, Czech Republic

**Keywords:** monkeypox virus, orthopoxvirus, poxviridae, outbreak, infection, zoonosis

## Abstract

Herein, we present our findings of an early appearance of the Monkeypox virus in Prague, Czech Republic. A retrospective analysis of biological samples, carried out on the 28th of April, revealed a previously unrecognized case of Monkeypox virus (MPxV) infection. Subsequent data analysis confirmed that the virus strain belongs to the ongoing outbreak. Combined with clinical and epidemiological investigations, we extended the roots of the current outbreak at least back to 16th of April, 2022.

## 1. Introduction

Monkeypox is a zoonotic infectious disease, closely related to smallpox and other Orthopoxviruses, with a clinical presentation similar to smallpox [1]. While earlier studies reported that human-to-human transmission is rare, recent studies reported high attack rates and thus possible outbreak/pandemic potential [2,3]. However, the fatality rate of monkeypox is several times lower than smallpox [4]. Moreover, it seems that people vaccinated against smallpox tolerate the infection better compared to the naive population (e.g., [1,2,5]). On the other hand, only the higher-age classes of the population went through smallpox vaccination at least in Europe, where smallpox was successfully eradicated [6]. This situation was addressed through the rapid acquisition of smallpox vaccines in the United Kingdom that will be primarily offered to close contacts of those diagnosed with monkeypox, as well as to health workers who will care for such patients [5].

According to the surveillance summary released by European Centre for Disease Prevention and Control (ECDC) up to 2nd of August 2022, a total of 15,926 MPxV cases have been identified in 38 countries throughout the Europe [7]; moreover, the ongoing monkeypox outbreak was declared a Public Health Emergency of International Concern by the WHO [8]. The epidemiological situation in the Czech Republic considering the MPxV outbreak seems to be under control, since up to date there have been only 25 confirmed cases. To argue whether it is caused by timely diagnostics and effective epidemiologic measures or simply by chance is beyond the scope of this article.

Here, we present a detailed description of an early case, confirmed both clinically as well as by Whole-Genome Sequencing (WGS) data and analyses. This report illustrates one of the earliest well-documented cases of the current outbreak.

## 2. Monkeypox Detection in the Czech Republic

In Europe, Monkeypox was recognized as an emerging threat in the first half of May 2022, when Monkeypox cases were reported in Portugal and the United Kingdom [9,10]; these people had no history of traveling to Africa where the virus is endemic [11].

In the Czech Republic, the first laboratory confirmed case was reported on 24th of May 2022 [12]. The patient was examined in the Military University Hospital in Prague, Department of Infectious Diseases. With regard to the clinical symptoms and travel history suggesting a risk contact (festival in Antwerp [13]) and numerous cases already reported in Europe, the possibility of Monkeypox was considered.

Surprisingly, through standard epidemiological investigation, it turned out that this patient was in contact with a patient hospitalized in Prague who was displaying similar symptoms from the second half of April (detailed clinical report in the Appendix A), i.e., before the first cases were reported from the United Kingdom. Biological samples collected from the patient were then retrospectively analyzed to confirm/decline the possibility of unrecognized MPxV infection.

## 3. Laboratory and Molecular Investigations

At the time of reinvestigation, the original anal swab sample was available in two forms, i.e., isolated DNA and the remaining swab in viral transport media. In these samples, we confirmed the presence of Orthopoxvirus by Transmission Electron Microscopy (TEM) (see Figure 1), which was subsequently determined as MPxV by real-time PCR according to Scaramozzino et al. [14] (detailed information concerning the used methods can be found in the Appendix A).

Furthermore, we sequenced the genome of the sample using the Oxford Nanopore Technologies (ONT) GridION platform in order to either confirm or decline that it belongs to the current outbreak. Nevertheless, this task turned out to be non-trivial, especially considering that the former patient was dismissed roughly a month before the time of the analysis. Therefore, the amount and quality of biologically relevant material was limited.

Firstly, we directly sequenced the original DNA sample (also used as a template for PCR). Despite quite deep sequencing, only a minor fraction of reads (up to ~0.03%) originated from the target (viral) genome. Moreover, poor quality and comparatively short reads suggested the degradation and fragmentation of the viral genome within the originally isolated DNA sample. Therefore, the data were sufficient only for the confirmation of the presence of MPxV.

In parallel, we applied an amplicon tiling strategy that could be compared to the COVID-19 sequencing protocol [15]. However, the technique combines 36 unique primer pairs and was not yet optimized. Therefore, the acquired data covered only roughly 75% of the virus genome, which was still not an optimal result.

Finally, we succeeded in the cultivation of the virus from the original swab sample, which allowed the isolation of a sufficient amount of viral DNA in the required quality for the ONT sequencing, i.e., high-molecular-weight DNA. As a result, we sequenced the whole genome of the virus with sufficient coverage (~350×).

Nevertheless, it should be noted that passaging the viruses might result in their adaptation to such artificial environments. Such a behavior was described even for poxviruses, however, under very specific conditions [16]. Moreover, our sample went only through one passage, which further minimizes the possibility of mutations.

Phylogenetic analysis performed using the assembled genome of the sample confirmed its clear association with the current outbreak (see Figure 2). The genome presented here is basically indistinguishable from the others of the ongoing outbreak, which might indicate stability of the lineage in its current environment/host or represent a typical behavior of slow-evolving dsDNA virus whereas some reports suggested that the Monkeypox lineage now circulating in Europe experienced, at least for some period, faster/rapid evolution, i.e., accumulated higher number of mutations than could be expected [17].

## 4. Discussion

The COVID-19 pandemic has represented an unprecedented crisis for modern mankind which has affected nearly all dimensions of society and our usual daily lives [18,19,20]. Nevertheless, it should be noted that the pandemic led to massive deployment and adaptation of Whole-Genome Sequencing (WGS) techniques in diagnostics and clinical practice, which might, if applied in a timely and effectively manner and in combination with the help of specialists from the field of molecular biology, lead to better risk assessment and mitigation [21]. WGS represents a fundamental tool for evolutionary biology as well as epidemiology. It is the only tool that allows the reliable tracking of the onset and spread of individual variants/strains and subsequently to assess their specific clinical manifestations [22,23,24].

We identified and documented a new early case of Monkeypox virus that modifies the comprehension of the current outbreak and its timeline. Considering the epidemiological history of the patient, the first symptoms of the disease were manifested already on 24th of April 2022. Furthermore, he claimed to have had unprotected sexual contact with a male of unknown nationality, Caucasian type, on 16th of April in Lisbon, Portugal. This suggests that the virus has been circulating in Europe at least from the first half of April 2022 (see Figure 3).

This study documented the possibility that before the Monkeypox outbreak was recognized, some early cases might still be awaiting the correct diagnosis and description. Therefore, we encourage for re-analysis of suspicious samples/cases where possible.

## 5. Conclusions

This study highlighted the importance of the cooperation of clinicians and epidemiologists with scientific institutions not only for the timely discovery of emerging outbreaks, but most importantly for discovering their roots. While timely diagnostics might prevent the spread of an outbreak within a country, studying its roots might lead to a better understanding of its dynamics and mitigation.

Based on combined evidence from clinical, epidemiological and molecular data, we report an early case of the Monkeypox virus from the Czech Republic. We evidenced that the virus responsible for the current outbreak was circulating within the European population before 16th of April 2022.

## Figures and Tables

**Figure 1 viruses-14-01773-f001:**
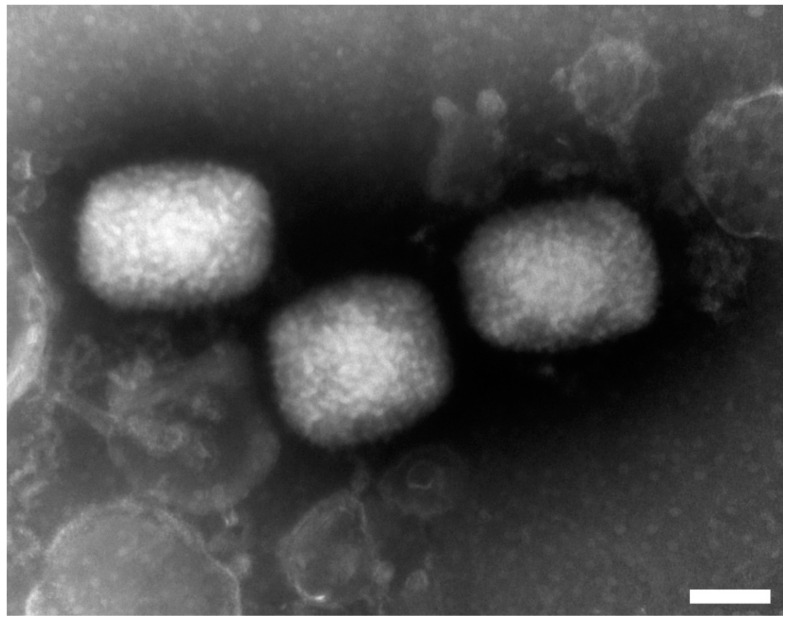
Orthopoxvirus particles. Transmission Electron Microscopy. magnification—150,000×. The length of the white bar in the right bottom corner represents 100 nm.

**Figure 2 viruses-14-01773-f002:**
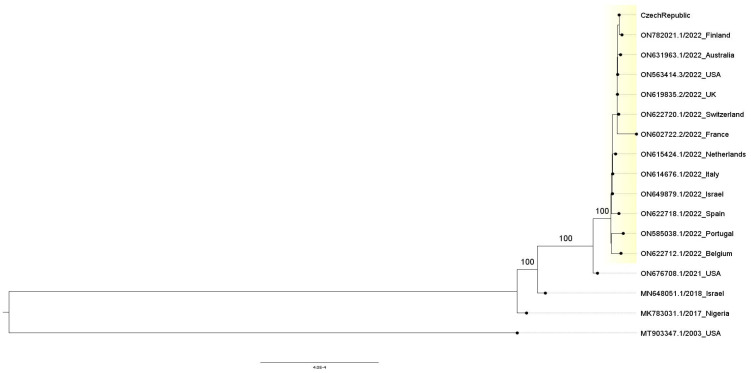
Maximum-likelihood phylogenetic tree: samples presumably originating from the current Monkeypox outbreak are highlighted; only (relevant) bootstrap values >85 were reported.

**Figure 3 viruses-14-01773-f003:**
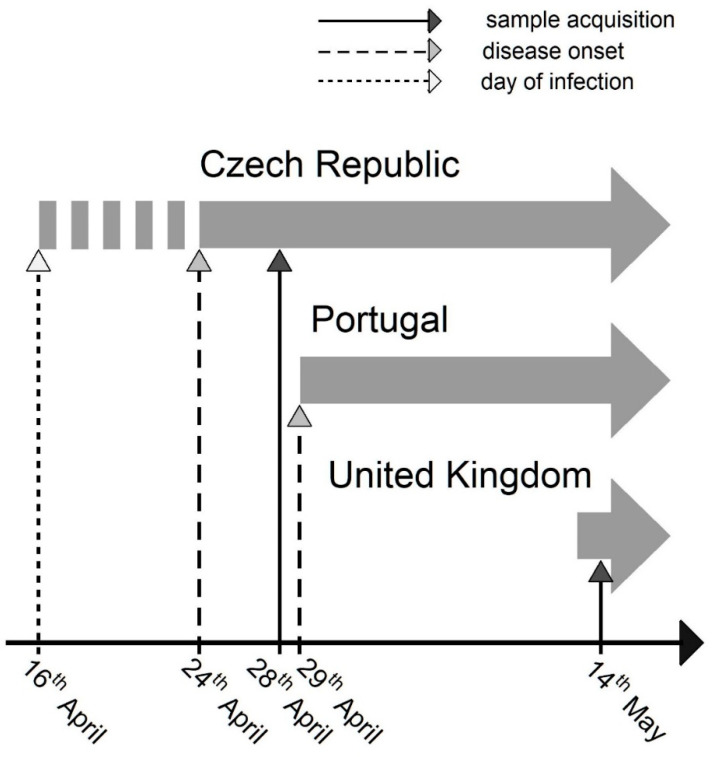
Timeline of the earliest reported Monkeypox virus infections from the European outbreak; only cases with no direct link to Africa were considered. Concerning the case from the Czech Republic, we provide three different dates: “sample acquisition” refers to a date when biological sample was acquired for molecular analyses; “disease onset” refers to a date when first symptoms of the disease were manifested; “day of infection” refers to a date when the infection was presumably transmitted (risk contact). Concerning the Portugal and United Kingdom cases, we report dates that were publicly available [9,10].

## Data Availability

Data are available upon request.

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
