# Peer review of "Retrospective Analysis Revealed an April Occurrence of Monkeypox in the Czech Republic"

_viruses, 2022, doi:10.3390/v14081773_

Round 1

Reviewer 1 Report

Seeking the roots of Monkeypox outbreak: retrospective analysis revealed an April occurrence in the Czech Republic by Chmel et al

The manuscript of this group is brief article investigating one of the first well-documented cases of the current MPXV outbreak and is therefore worthy of publication. It combines epidemiological data with electron microscopy and next-generation-sequencing. A small but sufficient phylogenetic assignment confirms the sample to belong to the current ongoing outbreak.

Nevertheless, I suggest the editor as well the authors to address some points that are easy and fast to implement before publication.

11.)    Title: “Seeking the roots of Monkeypox outbreak” is definitively an eyecatcher, but is not answered with the content of this manuscript. Please tune down and just implement “Monkeypox” into the second half of the title.

22.)    Introduction: Please adjust current case numbers before re-submitting to keep the manuscript up-to-date. Please add the information, that WHO Director-General declared the ongoing monkeypox outbreak a Public Health Emergency of International Concern on July 23rd. (https://www.who.int/europe/news/item/23-07-2022-who-director-general-declares-the-ongoing-monkeypox-outbreak-a-public-health-event-of-international-concern)

33.)    Supplemental data: Please move the case-presentation of the second case to the introduction of the main text body. This is essential, as this is the case is because of the epidemiological data shows the possible introduction in the second half of April into the Czech Republic. As the text there is well written as well it is just copy & paste with adding one or two linking sentences.

44.)    Discussion: Add a small paragraph dealing with “possible early unrecognized cases in other countries of Europe” combined with the need of increasing awareness for patients of risk and medical professionals.   

Author Response

Dear reviewer,
thank you very much for your comments. Please see the attached file (Rebuttal letter) with our responses to your remarks.
Sincerely
Martin Chmel

Reviewer 2 Report

Chmel et. al. report a very important finding to determine the earliest known case of Monkeypox  virus (MPVX) from the current outbreak. They find a case as early as Late April in Czech Republic. It's a very interesting study as determination of potential root of the current outbreak of MPVX will provide future preparedness for mitigation strategies. Even though the study is super important and needs to be published, I have few concerns about the rigor of the results. 

Following are my concerns:
1. TEM image of monkeypox should be verified by some other ways. Either Immuno EM assay or analyzing the image for size of poxvirus (~200-300 nm) with dumbbell shaped core visible. Please include how many panels were analyzed for TEM image and how can you be sure this is monkeypox in discussion? 

2. Please include your qRT-PCR data in main manuscript. Did you quantify virus particle via qRT-PCR or used regular PCR to amplify 14KD gene (please use gene name)? Provide these information.  

3. Authors couldn't get a good quality DNA from biological sample but were able to get infectious particle to amplify in cell culture to perform WGS. Poxviruses have shown to adapt (via mutation) in cell culture in less than 5X cycle, so please include the information on how many cycle virus was passaged in cells. Also provide limitation of this technique to let readers understand the nuances. 

4. The paragraph about WGS sequencing is not required for this manuscript, rather please explain the limitation of this study and potential benefit to the scientific community. Also explain about the importance of molecular tools for epidemiological study. 

Author Response

(The authors gave the same response as above.)

Round 2

Reviewer 2 Report

Thank you authors for providing more information. I would recommend it to be published at the current state.